# Comparing Monte Carlo simulations, mean particle theory estimates, and observations of $H^+$ and $O^+$ outflows at high altitudes and latitudes.

Imad A. Barghouthi[1], May R. Halaika[1]

[1]Department of physics, Al-Quds University, Jerusalem, Palestine.

*Correspondence to*: Imad A. Barghouthi (barghouthi@staff.alquds.edu)

**Abstract.** We conducted a comparative analysis of the results from Monte Carlo simulations, mean particle theory estimates, and available observational data across different regions of Earth's magnetosphere, including the auroral, polar wind, central polar cap, and cusp regions, focusing on the outflow of $H^+$ and $O^+$ ions at high latitudes and altitudes. We present altitude profiles for the mean perpendicular energy $W_\perp$, mean parallel energy $W_\parallel$, and mean total energy $W_{total}$. The Monte Carlo simulations were carried out using the Barghouthi model [Barghouthi, 2008], while the mean particle theory estimates were derived from Chang et al. [1986], and the observational data were obtained from Nilsson et al. [2013] and Barghouthi et al. [2016]. The results of the comparison across different regions reveal the following findings: 1) Monte Carlo simulations and mean particle theory yield similar results in the auroral regions but show discrepancies in the polar wind region. This discrepancy is attributed to the dominance of wave-particle interactions, which overshadow the effects of external forces (such as gravity, the polarization electric field, the mirror force, and centrifugal acceleration) in the auroral region and compete with them in the polar wind region. 2) The use of altitude-dependent diffusion coefficients leads to the generation of high-energy particles, which are not consistent with the corresponding observational data at middle and high altitudes. Therefore, we recommend the use of velocity and altitude-dependent diffusion coefficients. 3) A comparison with observations in the auroral and polar wind regions demonstrates excellent agreement in the auroral region and good agreement in the polar wind region, attributed to the implementation of appropriate velocity and altitude-dependent diffusion coefficients. 4) In the central polar cap and cusp regions, both methods and observations show excellent agreement. 5) Based on these comparisons, we conclude that the electromagnetic wave wavelength in the polar wind and auroral regions is approximately 8 km, and the velocity and altitude-dependent diffusion coefficients used in the Monte Carlo simulations and mean particle theory are suitable for application in further studies of these regions.

## 1 Introduction

Numerous studies, including analytical, modelling, and observational approaches, have been conducted to investigate the behaviour of ion outflow (e.g., $O^+$ and $H^+$) from Earth's polar regions into outer space. These studies aim to characterize various properties of the ion outflow, such as velocity distribution, temperature, density, drift velocity, and heat flux. Ongoing

research efforts continue to enhance our understanding of the dynamics of these ion flows, with particular focus on the energy and its components (parallel, perpendicular, and total energy), which constitutes the primary subject of this study.

Various researchers have investigated ion outflow at high altitudes and latitudes. Chang et al. [1986] introduced the mean particle theory, which explains the perpendicular heating of ions in a dipole magnetic field. They suggested that intense broadband electric field fluctuations observed in the frequency range of 0 -100 Hz could induce transverse ion activation through cyclonic resonance heating by left-handed polarized electromagnetic waves. Additionally, by employing a set of equations governing ion motion within the geomagnetic field, they derived expressions for the parallel and perpendicular energies based on the mean particle theory. Retterer et al. [1987] demonstrated how oxygen ions form conic distributions in the auroral zone and utilized the diffusion equation to explain ion velocity distributions obtained through the Monte Carlo method.

Barakat and Barghouthi [1994a, 1994b] enhanced the Monte Carlo simulation and explored the effects of wave-particle interactions (WPI) on $H^+$ and $O^+$ ion outflow in the polar wind. Their model incorporated the electrostatic field, gravity, and geomagnetic field lines. These studies are considered parametric investigations as they used constant values for the quasilinear velocity diffusion rates along the simulation tube. The velocity distribution function and its corresponding velocity moment profiles were simulated and presented for both ion species. Barghouthi [1997] and Barghouthi et al. [1998] analyzed data from the Plasma Wave Instrument (PWI) aboard the Space Dynamics Explorer 1, calculating the altitude dependence of the velocity diffusion rate. They examined the impact of altitude-dependent wave-particle interactions (WPI) on $H^+$ and $O^+$ ion outflow in both the polar cap and auroral zones using Monte Carlo simulations. Additionally, Barghouthi [1997] compared energy estimates from the mean particle theory (Chang et al., [1986]) with the corresponding energy results produced by Monte Carlo simulations in the auroral region. Despite the lack of supporting observational data, strong agreement was found between the Monte Carlo simulations and the mean particle theory estimates.

Bouhram et al. [2002, 2003a, 2003b, 2004] developed a two-dimensional Monte Carlo model to investigate ion outflow from the dayside cusp/cleft, focusing on the effects of transverse ion heating. They examined the mechanisms of transverse heating and ion outflow within the cusp/cleft region. Using their model, they interpreted Cluster satellite observations, specifically the saturation of the transverse energization rate, in terms of the influence of finite perpendicular wavelength effects in wave-particle interactions.

Barghouthi and Atout [2006] focused on Monte Carlo simulations of toroidal $H^+$ and $O^+$ velocity distributions at high altitudes, equatorward of the cusp, by employing an appropriate form for the velocity diffusion coefficient $D_\perp$. The results of the Monte Carlo simulations, including the toroidal $H^+$ and $O^+$ velocity distributions as well as the $H^+$ and $O^+$ ion temperatures, were compared to the corresponding toroidal $H^+$ and $O^+$ ion distributions and ion temperatures observed at high altitudes, equatorward of the cusp [Huddleston et al., 2000]. These comparisons yielded reasonable agreement.

Barghouthi [2008] utilized Monte Carlo simulations to determine the temperatures and velocity distributions of $H^+$ and $O^+$ ions at high altitudes in the equatorward portion of the cusp, using various forms of altitude- and velocity-dependent diffusion

coefficients $D_\perp(r, v_\perp)$, including the RCC model, Bouhram model, and Barghouthi model. The simulation results were compared with the corresponding observations from Huddleston et al. [2000], and the results from the Barghouthi model showed excellent agreement with the observations. Furthermore, Barghouthi [2008] provided substantial evidence, including comparisons between Monte Carlo simulation results for $H^+$ and $O^+$ ion outflows obtained using the Barghouthi model and corresponding observational data from multiple sources at various altitudes in the auroral region, further supporting the validity

of the Barghouthi model.

    Waara et al. [2010] presented a case study of significant heating of outflowing oxygen ions at high altitudes (12 $R_E$) above the polar cap, with energies reaching up to 8 keV perpendicular to the geomagnetic field. The shape of the distribution functions suggests that the majority of the heating occurs locally, within 0.2–0.4 $R_E$ of altitude. They found that the locally observed wave fields are unlikely to account for the observed ion energization. Additionally, it is improbable that the ions originated

from a nearby energizing location and migrated to the observation site. These findings indicate the existence of additional, fundamentally distinct ion energization mechanisms at high altitudes. One possible explanation is that the ions' magnetic moment is not conserved, which would result in slower outflow velocities and an extended period of ion energization.

    Waara et al. [2011, 2012] conducted a statistical analysis of ion heating and associated wave activity. They provided average values for coefficients that describe diffusion in ion velocity space at various altitudes, offering a useful framework for studying

ion outflow behaviour and energy characteristics. Their test particle calculations suggest that the average energies of $O^+$ ions correlate with the observed wave activity at high altitudes (8-15 $R_E$) in the cusp and mantle regions. They also found that the electric-to-magnetic field spectral density ratios closely match the expected values for Alfven waves. According to their findings, the diffusion coefficient for $O^+$ ions increases with altitude.

    Barghouthi et al. [2012] compared simulation results of ion outflow (including ion density, drift velocity, perpendicular and

parallel temperatures, and ion velocity distributions at various altitudes) in two distinct regions, the polar wind and auroral regions, based on the Barghouthi model. They found that wave-particle interactions had a more significant effect in the auroral zone compared to the polar wind region and that these interactions had a greater influence on the energization of $O^+$ ions than on $H^+$ ions.

    Barghouthi et al. [2016] updated the Monte Carlo model to include the effects of gravity, ambipolar electric field, centrifugal

acceleration, mirror force, and wave-particle interactions in the study of $O^+$ and $H^+$ ion outflow above the polar cap. They modified several parameters, such as centrifugal acceleration, velocity diffusion coefficients, and boundary conditions at lower altitudes. The results were compared with observational data obtained from various instruments aboard the Cluster spacecraft, and the simulation results were found to be in good agreement with the observed data.

    The primary objective of this study is to compare the simulation results (perpendicular energies $W_\perp$, parallel energies $W_\parallel$, and

total energies $W_{total}$) of $O^+$ and $H^+$ ions obtained using the Monte Carlo model (Barghouthi model) and mean particle theory with available observational data from various regions of Earth's magnetosphere, including the polar wind, auroral region,

cusp, and central polar cap. This comparison seeks to evaluate the importance of including altitude-dependent velocity diffusion coefficients $D_\perp(r)$ or altitude-and velocity-dependent velocity diffusion coefficients $D_\perp(r, v_\perp)$, while also considering the constraints posed by wavelength limitations.


## 2 Formulations

### 2.1 Monte Carlo simulation

In the study of space plasma, it is advantageous to describe each constituent species utilizing distinct velocity distribution function, denoted as $f_s(\mathbf{v}_s, \mathbf{r}_s, t)$. The velocity distribution function is defined such that the expression $f_s(\mathbf{v}_s, \mathbf{r}_s, t) d\mathbf{v}_s d\mathbf{r}_s$

quantifies the number of particles of species $s$ that, at time $t$, possess velocities within the interval $\mathbf{v}_s$ and $\mathbf{v}_s + d\mathbf{v}_s$ and positions between $\mathbf{r}_s$ and $\mathbf{r}_s + d\mathbf{r}_s$. The temporal evolution of the species' velocity distribution function is governed by the cumulative effects of collisions and interactions, as well as the dynamical movements of the species in phase space under the influence of external forces [Schunk, 1977]. This evolution can be mathematically represented by the well-established Boltzmann equation:

$$\frac{\partial f_s}{\partial t} + \mathbf{v}_s . \nabla f_s + \left(\frac{e_s}{m_s}\right)\left[\boldsymbol{E} + \frac{\mathbf{v_s \times B}}{c}\right] . \nabla_{v_s} f_s = \frac{\delta f_s}{\delta t} \qquad (1)$$

In this equation, the left-hand side represents the evolution of the velocity distribution function $f_s(\mathbf{v}_s, \mathbf{r}_s, t)$ under the effects of external forces, and the right-hand side represents the Boltzmann collision integral, here it represents the rate at which $f_s$ changed as a result of wave particle interactions in the region of study. In the above equation, $E$ is the polarization electric field, $B$ is the geomagnetic field, $c$ is the speed of light, $\nabla$ is the coordinate space gradient, and $\nabla_{v_s}$ is the velocity space gradient, and $e_s$ and $m_s$ are the charge and mass of species $s$, respectively. The suitable expression for $\left(\frac{\delta f_s}{\delta t}\right)$ in case of wave

particle interactions is given by Retterer et al. [1987], they considered the effects of (WPI) as particles diffusion in the velocity space.

$$\left.\frac{\delta f}{\delta t}\right|_{WPI} = \left(\frac{1}{v_\perp}\right)\frac{\partial}{\partial v_\perp}\left[D_\perp v_\perp \frac{\partial f}{\partial v_\perp}\right] \qquad (2)$$

where $D_\perp$ is provided by Retterer et al. [1987] and represents the quasi-linear velocity diffusion rate perpendicular to the geomagnetic field,

$D_\perp = (\eta q^2/4m^2)|E_x(\omega = \Omega)|^2 \qquad (3)$

where $|E_x(\omega)|^2$ is the measured spectral density of the electromagnetic turbulence, $\eta$ is the proportion of the measured spectral density by plasma wave instrument (PWI) on board dynamic explorer 1 (DE-1) spacecraft that corresponds to the left-hand polarized wave, $q$ is the ion's charge, $m$ is the ion's mass, $\Omega$ is the ion's gyrofrequency, and $\omega$ is the angular frequency of the electromagnetic turbulence.

The expression of the velocity diffusion rate $D_\perp$ as given in Eq. (3) is independent of velocity and depends on position (altitude) via changes in the ion gyrofrequency, $\Omega$, along the geomagnetic field lines.  By examining experimental data of electric field spectral density obtained by plasma wave instrument (PWI) onboard the DE-1 satellite (i.e. for high solar activity conditions),

Barghouthi [1997] and Barghouti et al. [1998] calculated the altitude dependence of ($D_\perp$). They came up with the following expressions for the velocity diffusion coefficient $D_\perp$ in the polar wind region [Barghouthi et al., 1998] as follows:


$$D_\perp (r) = \begin{cases} 5.77 \times 10^3 \left(\frac{r}{R_E}\right)^{7.95} cm^2 s^{-3}, \text{for } H^+ \\ 9.55 \times 10^2 \left(\frac{r}{R_E}\right)^{13.3} cm^2 s^{-3}, \text{for } O^+ \end{cases} \tag{4}$$

In the auroral region, $D_\perp (r)$ is given by Barghouthi [1997] as follows:

$$D_\perp (r) = \begin{cases} 4.45 \times 10^7 \left(\frac{r}{R_E}\right)^{7.95} cm^2 s^{-3}, \text{for } H^+ \\ 6.94 \times 10^5 \left(\frac{r}{R_E}\right)^{13.3} cm^2 s^{-3}, \text{for } O^+ \end{cases} \tag{5}$$

In central polar cap (CPC) and cusp regions, $D_\perp (r)$ is given by Nilsson et al. [2013] as follows:

For central polar cape region

$$D_\perp (r) = \begin{cases} 20 \left(\frac{r}{R_E}\right)^{9.77} cm^2 s^{-3}, \text{for } H^+ \\ 0.5 \times 10^5 \left(\frac{r}{R_E}\right)^{5.5} cm^2 s^{-3}, \text{for } O^+ \end{cases} \tag{6}$$

and for cusp region

$$D_\perp (r) = \begin{cases} 1.01 \times 10^6 \left(\frac{r}{R_E}\right)^{5.61} cm^2 s^{-3}, \text{for } H^+ \\ 2.5 \times 10^4 \left(\frac{r}{R_E}\right)^{6.4} cm^2 s^{-3}, \text{for } O^+ \end{cases} \tag{7}$$

The diffusion coefficient was given a new form by Barghouthi [2008], who discovered that it is a velocity-dependent in
addition to altitude-dependent.

$$D_\perp (r, v_\perp) = D_\perp (r) \begin{cases} 1 & \text{for } \left(\frac{k_\perp v_\perp}{\Omega_i}\right) < 1 \\ \left(\frac{k_\perp v_\perp}{\Omega_i}\right)^{-3} & \text{for } \left(\frac{k_\perp v_\perp}{\Omega_i}\right) \geq 1 \end{cases} \tag{8}$$

Where $D_\perp (r, v_\perp)$ is the quasi-linear velocity diffusion rate perpendicular to the geomagnetic field lines (altitude and velocity dependent), $\Omega_i$ is the ion gyrofrequency and $k_\perp$ is perpendicular wave number and related to the characteristic perpendicular wavelength of the electromagnetic turbulence $\lambda_\perp$. Equation (8) indicates the diffusion coefficient that is dependent on altitude
and velocity. However, as ions are heated and move to higher altitudes, their gyroradius may approach the perpendicular wavelength of the electromagnetic turbulence, and when the ratio ($k_\perp v_\perp / \Omega_i$) exceeds 1, the heating rate becomes self-limited. Bouhram et al. [2004] derived an alternative form for the altitude and velocity dependent diffusion coefficient and interpreted their results in terms of finite wavelength effects.

By solving Boltzmann equation, Eq. (1), using Monte Carlo technique the velocity distribution functions were obtained for
each species (in this study $O^+$ and $H^+$ ions) and its velocity moments, i.e. density $n_s$ , drift velocity $u_s$, and parallel $T_{s\parallel}$ and perpendicular $T_{s\perp}$ temperatures. The moments considered here are defined as follows [Barghouthi, 1997]:

$$n_s = \int f_s d\mathbf{v}_s \qquad\qquad (9)$$

$$u_s = \frac{1}{n_s} \int v_{s\parallel} f_s d\mathbf{v}_s \qquad\qquad (10)$$

$$T_{s\parallel} = \frac{m_s}{n_s k} \int \left(v_{s\parallel} - u_s\right)^2 f_s d\mathbf{v}_s \qquad\qquad (11)$$

$$T_{s\perp} = \frac{m_s}{2 n_s k} \int (v_{s\perp})^2 f_s d\mathbf{v}_s \qquad\qquad (12)$$

These Monte Carlo results will be used to calculate the mean parallel energy, mean perpendicular energy, and total mean energy as given in the following expressions [Barghouthi, 1997], respectively:

$$W_{s\parallel} = \frac{1}{2} m u_s{}^2 + \frac{1}{2} k T_{s\parallel} \qquad\qquad (13)$$

$$W_{s\perp} = k T_{s\perp} \qquad\qquad (14)$$

$$W_s = W_{s\parallel} + W_{s\perp} \qquad\qquad (15)$$

Where $u_s$, $T_{s\parallel}$ and $T_{s\perp}$ are given by equations (10), (11) and (12), respectively and $W_{s\parallel}$ and $W_{s\perp}$ are the mean parallel and perpendicular energies, respectively; $W_s$ is the total mean energy; and s denotes the type of the ion ($O^+$ or $H^+$), $k$ is Boltzmann constant.

## 2.2 Barghouthi model

The Barghouthi model was developed to investigate the behavior of ion outflows, specifically for $H^+$ and $O^+$ ions, at high altitudes and high latitudes. The simulation results generated by this model show excellent agreement with observational data from various regions, including the auroral region [Barghouthi, 2008] and the polar wind region [Barghouthi et al., 2011]. This model accounts for multiple influential factors, including gravity, the polarization electric field, the diverging geomagnetic field, and wave-particle interactions, all of which affect $H^+$ and $O^+$ ion outflows at elevated altitudes and latitudes. Notably,

the model highlights the significant contribution of wave-particle interactions, which are responsible for ion heating. The impact of these wave-particle interactions is characterized by the velocity diffusion coefficient $D_\perp (r, v_\perp)$, which has been formulated as a function of position $(r/R_E)$ along Earth's geomagnetic field lines and the perpendicular velocity of the injected ions $(v_\perp)$. Various functional forms of the velocity diffusion coefficient $D_\perp (r, v_\perp)$ have been employed in Monte Carlo simulations to derive the density, drift velocity, parallel and perpendicular temperatures, heat fluxes, and velocity distribution

functions for $H^+$ and $O^+$ ions at high altitudes and latitudes. In the Monte Carlo simulations of this study, the appropriate velocity diffusion coefficient corresponding to the specific study region, $D_\perp (r, v_\perp)$, was utilized to ascertain the temperatures and velocity distributions of $H^+$ and $O^+$ ions at these elevated altitudes and latitudes.

## 2.3 Mean particle theory

Chang et al. [1986] presented a theoretical framework for estimating the mean perpendicular, mean parallel, and total

mean energies as functions of geocentric distance. This framework incorporates the average heating rate for each ion into a set of equations that describes the motion of ions along geomagnetic field lines, as outlined below:

$$W_{i\parallel} = \frac{9m_i}{2^{1/3}} \left[ \frac{rD_\perp(r)}{(3\alpha+1)(6\alpha+11)} \right]^{2/3} \qquad (16)$$

$$W_{i\perp} = \frac{(6\alpha+2)m_i}{2^{1/3}} \left[ \frac{rD_\perp(r)}{(3\alpha+1)(6\alpha+11)} \right]^{2/3} \qquad (17)$$

$$W_i = \quad W_{i\parallel} + W_{i\perp} = (3\alpha + 11/2)^{1/3} m_i \left[ \frac{rD_\perp(r)}{(3\alpha+1)} \right]^{2/3} \qquad (18)$$

Where $W_\parallel$ and $W_\perp$ are the mean parallel and perpendicular energies, respectively, $W_i$ is the total mean energy, $i$ denotes the type of the ion ($H^+$ or $O^+$), and $\alpha$ is a fitting parameter. In that theory the mean energy ratio $W_\perp / W_\parallel$ asymptotically approaches a constant value.

## 3 Comparisons

In this section, we will compare Monte Carlo simulations with mean particle theory estimates in the auroral and polar wind

regions (sub-section 3.1). Additionally, we will present a comparison among Monte Carlo simulations, mean particle theory estimates, and relevant observational data in sub-section 3.2.

### 3.1 Comparison between Monte Carlo simulations and estimates of mean particle theory.

By employing the Monte Carlo technique (specifically the Barghouthi model) alongside mean particle theory, we have derived varying altitude profiles for the mean parallel, mean perpendicular, and total mean energies across different regions of Earth's

magnetosphere—specifically, the auroral region (Figure 1) and the polar wind region (Figure 2)—for $H^+$ (left panels) and $O^+$ (right panels) ions outflow. In both methods, we utilized an appropriate $D_\perp(r)$ as the velocity diffusion coefficient for each region. Overall, our findings in the auroral region (Figure 1) indicate an excellent agreement between


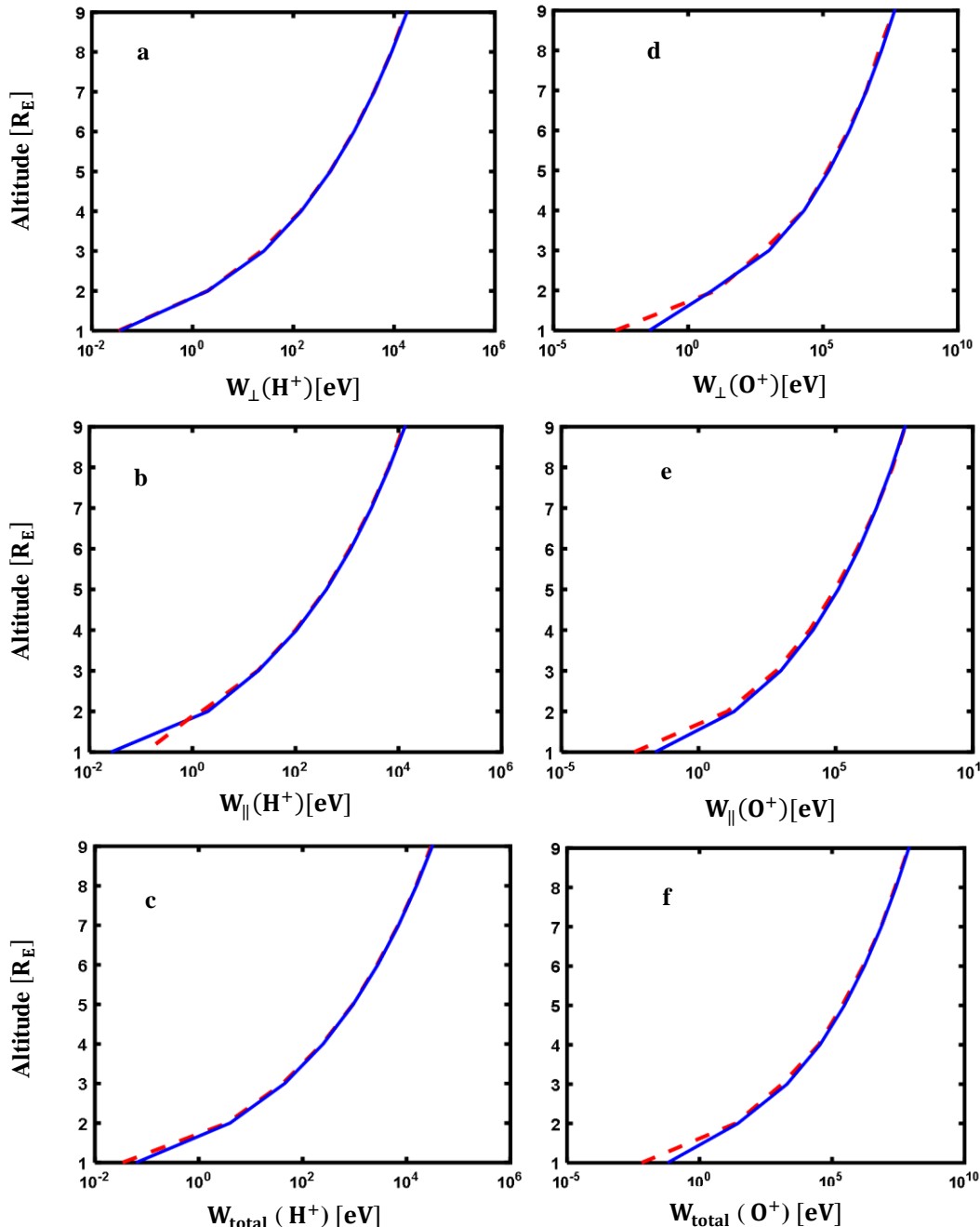

Figure 1: Altitude profiles comparing the mean particle theory estimates (blue solid lines) under auroral conditions with Monte Carlo calculations (red dashed lines). The left panels (a, b, and c) represent H+ ions, while the right panels (d, e, and f) illustrate O+ ions. Panels a and d show the mean perpendicular energy $W_\perp$, panels b and e present the mean parallel energy $W_\parallel$, and panels c and f depict the mean total energy $W_{total}$.

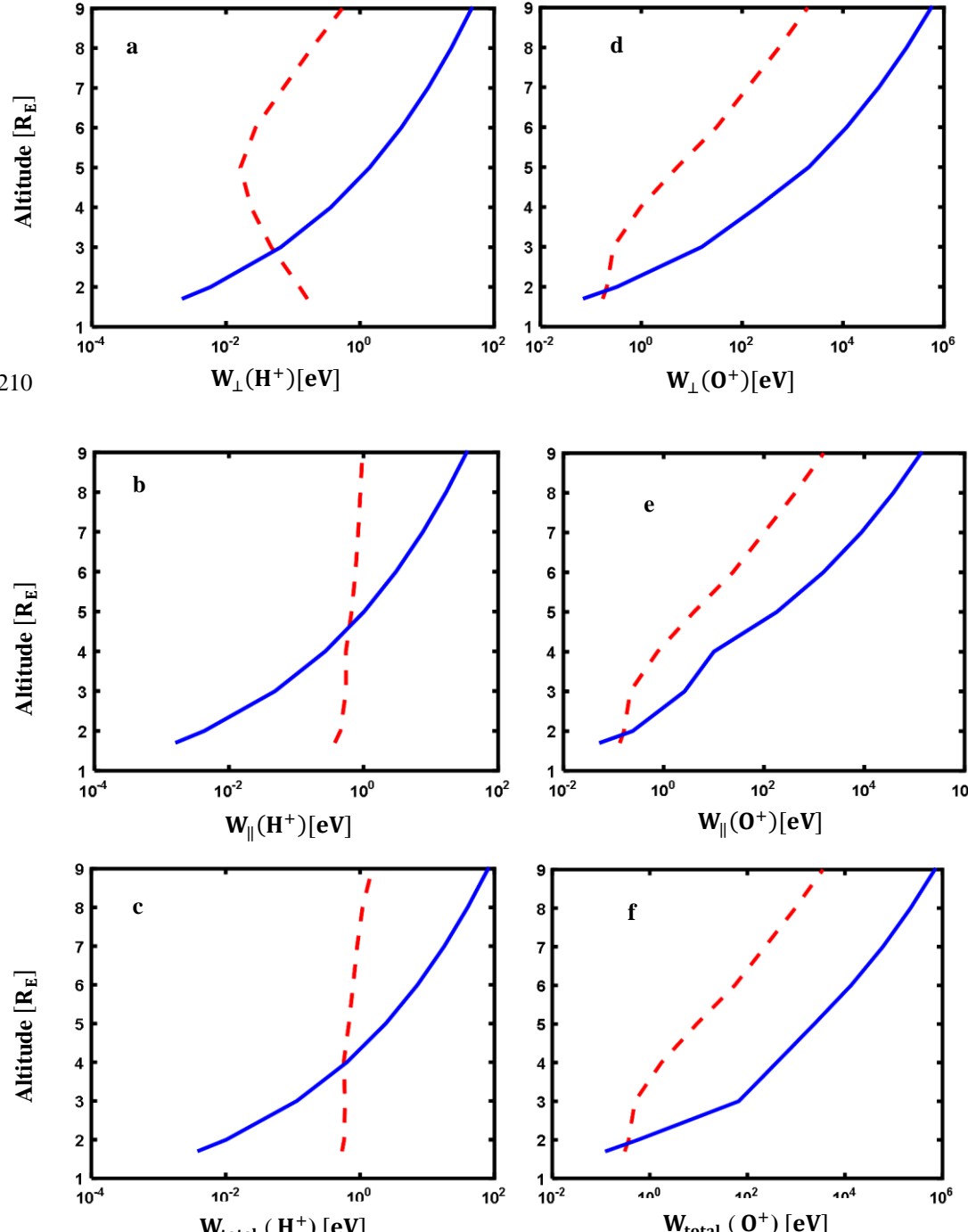


**Figure 2: Comparison of the estimates of the mean particle theory (blue solid lines) for polar wind conditions with the Monte Carlo calculations (red dashed lines). (right panels, d, e and f) for $O^+$ ions and (left panels, a, b and c) for $H^+$ ions and. The mean**
**perpendicular energy $W_\perp$ represented by (panels a and d), mean parallel energy $W_\parallel$ (panels b and e) and total energy (panels c and f).**

Monte Carlo simulations and estimates of mean particle theory for H$^+$ and O$^+$ ions. It is important to highlight that in the mean particle theory, ion heating and energization occur solely due to wave-particle interactions, which primarily depend on the value of the velocity diffusion coefficient. In contrast, the energization process in the Monte Carlo model results from the competition between wave-particle interactions and external forces, such as gravity, the polarization electric field, and the divergence of the geomagnetic field. The close agreement between the two methods can be attributed to the dominant influence of wave-particle interactions, which prevail over the effects of external forces due to the high values of $D_\perp(r)$. In other words, wave-particle interactions are the primary drivers of the energization process for both H$^+$ and O$^+$ ions in this auroral region. For polar wind region and for $\lambda_\perp \to \infty$, this means the velocity diffusion coefficient, $D_\perp(r)$, is altitude dependent. The values of the velocity diffusion coefficient are less than those in the auroral region, see equations (4) and (5). According to Fig. 2 there is no agreement between Monte Carlo simulations and mean particle theory estimates, this is due to the contribution of the external forces in Barghouthi model that competes with the effect of wave particle interaction, however in mean particle theory the external forces are not considered and the heating is due to wave particle interaction. We report here that mean particle theory is not suitable to be used in this region, and Monte Carlo simulations are more appropriate to be used as shown in Barghouthi et al. [2011] when they compared their Monte Carlo results with observations. Also, we have found that, when we use the diffusion coefficient that depends on altitude only, its value becomes very large as altitude increases. Therefore the values of the particle energies obtained from equations (16), (17) and (18) turn to be very high, but when the diffusion coefficient becomes velocity and altitude dependent according to eq. (8) (i.e. $k_\perp v_\perp / \Omega \geq 1$), the produced particle energies turn to be reasonable as shown in Fig.3 (blue solid lines and red dashed lines).

### 3.2 Comparison between Monte Carlo simulations, estimates of the mean particle theory, and available observations

In this sub-section, we present, only, the simulation results and the estimates of the mean particle theory that have corresponding observations. Barghouthi [2008] compared the Monte Carlo simulation results that obtained from Barghouthi model with the corresponding observations for H$^+$ and O$^+$ ions outflows in the auroral region at different altitudes in the simulation tube, he obtained an excellent agreement, particularly, when the typical perpendicular wavelength of the electromagnetic turbulence was 8 km. Also, he observed that there is a broad agreement between the simulation results of the polar wind for this wavelength and the corresponding observations. For these reasons, we chose to have the results of the comparison with corresponding observations when $\lambda_\perp = 8\ km$, i.e. when the velocity diffusion coefficient is altitude and velocity dependent. We will compare the outcomes of our Monte Carlo simulations, the estimates of mean particle theory, and available observations obtained from different published articles. Observations of O$^+$ ions at various altitudes were obtained for parallel velocity, perpendicular temperature and parallel temperature for both polar wind and auroral regions from [Nilsson et al., 2013] and observations for parallel velocity and perpendicular temperature for O$^+$ ions in central polar cap and cusp regions from Barghouthi et al. [2016]. In this study, we adopt the terms "good" and "excellent" agreement to qualitatively describe how well our simulation and theory curves correspond to observational data. These classifications are based on a

visual comparison with the observational ranges, represented by the maximum, average, and minimum values. We define
agreement as excellent when the simulation and theory curves largely overlap with the average observational curves and remain within the observed range across all altitudes. Conversely, we describe the agreement as good when the curves generally fall within the envelope formed by the observational maximum and minimum values, even if they partially deviate from the average trend. This qualitative approach is commonly used in comparative studies of ionospheric outflows (e.g., Barghouthi et al. 2011,2012, 2016), and it acknowledges the significant variability typically present in observational data.


Figure 3 displays the comparison results for the auroral (left panels) and polar wind (right panels) regions. It contrasts the Monte Carlo method (red dashed lines) and the estimates from mean particle theory (blue solid lines) at $\lambda_\perp = 8$ km, alongside observational data represented by maximum (dotted lines), average (dashed-dotted lines), and minimum (black dashed lines) values for $O^+$ ions. Across all four panels, it is evident that the results from the Monte Carlo simulations and the mean particle 260 theory estimates are closely aligned, demonstrating excellent agreement at lower altitudes. At higher altitudes, while both methods display similar qualitative behaviour and maintain good agreement, we attribute this acceptable and reasonable correlation to the application of a suitable altitude- and velocity-dependent diffusion coefficient in both approaches.


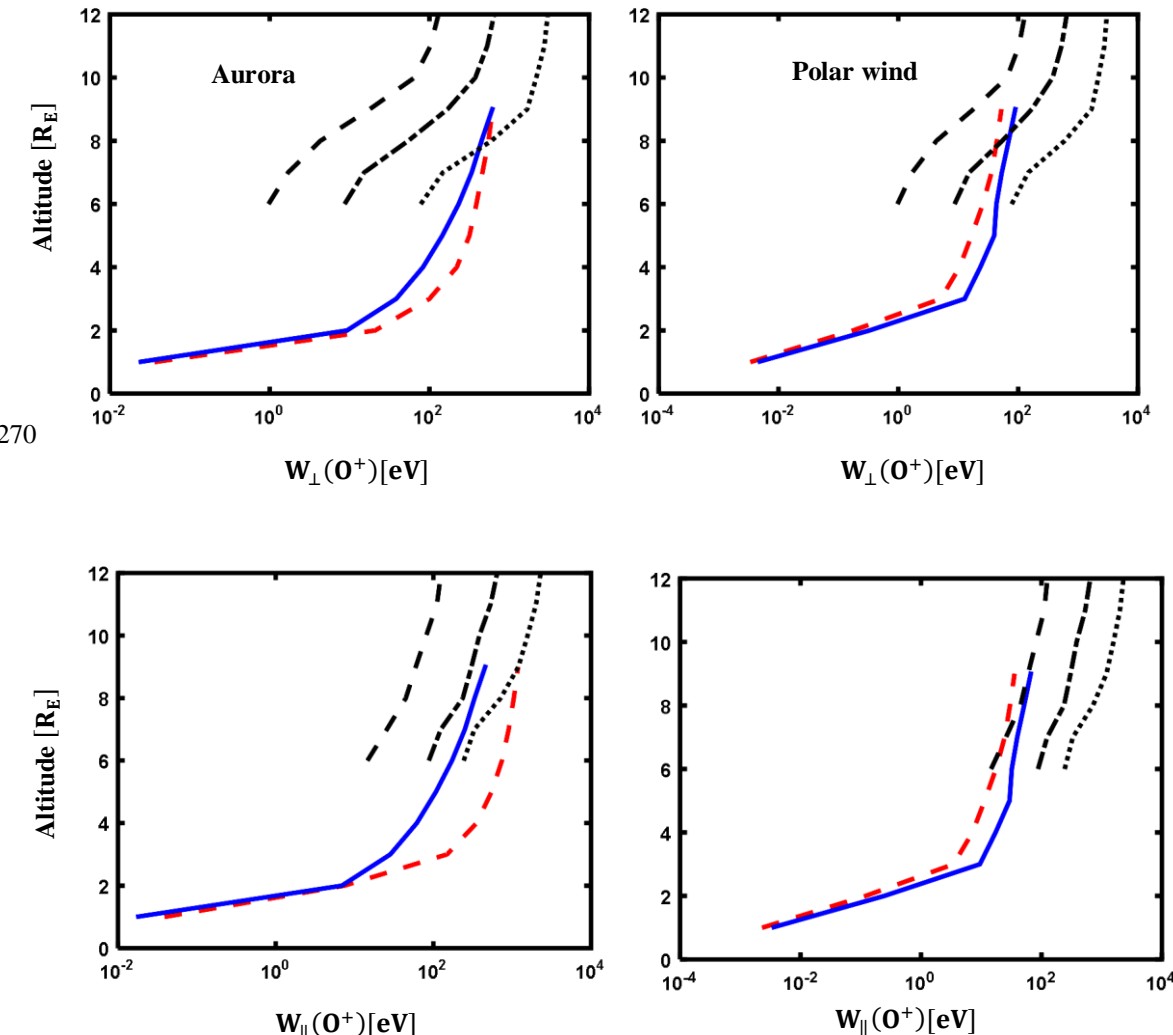

**Figure 3: Comparison of mean particle theory estimates (blue solid lines) and Monte Carlo calculations (red dashed lines) for auroral**
**conditions (left panels) and polar wind conditions (right panels), along with observational data represented by maximum (dotted lines), average (dashed-dotted lines), and minimum (black dashed lines) values for O+ ions. The top panel shows the mean perpendicular energy, while the bottom panel presents the mean parallel energy. In this analysis, we have considered the wavelength of electromagnetic turbulence to be $\lambda_\perp = 8$ km.**

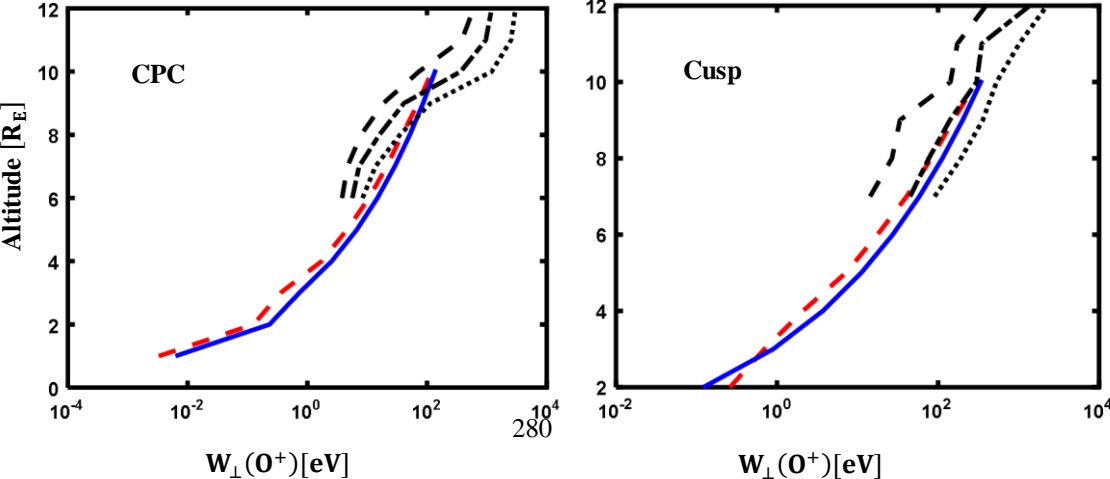

**Figure 4:** Comparison of the estimates of the mean particle theory (blue solid lines) with the Monte Carlo calculations (red dashed line), for central polar cap conditions (left panel) and cusp conditions (right panel) in addition to the observations (maximum (doted lines) , avarege (doted dashed lines), and minimum (black dashed lines)) for $O^+$ ions. In this analysis, we have considered the wavelength of electromagnetic turbulence $\lambda_\perp \rightarrow \infty$.

Based on the comparison with observations, it is evident that the simulation results and estimates from the mean particle theory fall within the observational range. This indicates that $D_\perp (r, v_\perp)$ is appropriate for use in both the Monte Carlo simulation and the mean particle theory. Specifically, the results for perpendicular energy $W_\perp (O^+)$ from both the Monte Carlo method and mean particle theory align well with the maximum observed values in the auroral region, and they also exhibit good agreement with the average observed values in the polar wind region. Regarding the mean parallel energy $W_\parallel (O^+)$ as shown in Fig. 3, bottom panels), the simulation results and estimates from the mean particle theory are closely aligned with both the average and maximum observed values in the auroral region, and they demonstrate excellent agreement with the minimum observed values in the polar wind region.

In the central polar cap (Fig. 4, left panel) and cusp (Fig. 4, right panel) regions, we utilized the altitude-dependent diffusion coefficient from Nilsson et al. [2013], specifically when $\lambda_\perp \rightarrow \infty$. This approach resulted in excellent agreement at all altitudes between the Monte Carlo simulations and the estimates of the mean particle theory. Furthermore, we observed a strong correspondence with observational data in both regions; the results closely match the maximum and average values in the central polar cap region and align well with the average values from observations in the cusp region.

As a result of these comparisons, it is crucial to select the appropriate form of the velocity diffusion coefficient for each region of interest within the Earth's magnetosphere when analysing ion energization and heating processes.

Barghouthi [2008] conducted a comparative study between Monte Carlo simulations—based on the Barghouthi model—and in-situ measurements of $O^+$ and $H^+$ ions at selected altitudes in the auroral region. This work was later extended by Barghouthi

et al. [2011] to include the polar wind region. In both studies, the simulation results sometimes showed good agreement with observational data in terms of energy magnitudes and general trends; however, notable discrepancies were also identified. These deviations were attributed to physical processes not incorporated in the model, such as variations in ion heating or cooling mechanisms, the influence of unmodeled external forces, or other dynamic plasma phenomena. Importantly, these earlier studies did not incorporate the mean particle theory, which emphasizes ion heating due to wave-particle interactions.


In the present study, both Monte Carlo simulation results and theoretical estimates based on mean particle theory were found to fall within the observational range, as shown in Figures 3 and 4. Specifically, we noted that in the top panels of Figure 3— covering altitudes between 6 and 8 Earth radii ($R_E$)—although the simulated ion energies align in magnitude with observed values, the altitude-dependent energy profiles differ significantly in trend. Theoretical and simulated profiles exhibit a steeper

gradient with altitude compared to the observational data. A similar behaviour can be seen for the left panel of Figure 4, where the theoretical profiles show steeper gradients than those observed, particularly at higher energies. In contrast, the right panel of Figure 4 demonstrates a stronger consistency between theoretical predictions and observations.

In our analysis, we examined both absolute energy values and the slopes of the energy profiles. The discrepancies noted in the

top panels of Figure 3—especially the steeper theoretical and simulated slopes—can be attributed to the use of altitude- and velocity-dependent diffusion coefficients that fully account for processes like ion self-limiting heating [Bouhram et al., 2004] and localized cooling effects. The deviations in the left panel of Figure 4 are mainly due to oversimplified assumptions regarding wave mode propagation and the absence of spatial variations in plasma conditions. By contrast, better agreement in the bottom panels of Figure 3 and the right panel of Figure 4 is achieved due to the implementation of empirically-based,

altitude- and velocity-dependent diffusion coefficients.

These findings underscore the importance of region-specific parameterizations and highlight the need for future improvements in the modelling framework. In particular, incorporating saturated wave-particle interactions, as well as multidimensional plasma dynamics, would enhance the model's ability to replicate the observed energy profiles more accurately.

**4 Conclusions**

We conducted a comparative analysis of energy components— mean perpendicular energy $W_\perp$, mean parallel energy $W_\parallel$, and total mean energy $W_{total}$, for $H^+$ and $O^+$ ions using both the Monte Carlo method and estimates from mean particle theory across various regions of Earth's magnetosphere, including the polar wind, auroral, central polar cap, and cusp regions. Utilizing altitude-dependent diffusion coefficients, we found excellent agreement between the two methods in the auroral

regions; however, there was a lack of concordance in the polar wind region. Moreover, we observed that energy values at middle and high altitudes were unrealistically elevated compared to observational data. To mitigate this issue, we implemented

velocity- and altitude-dependent diffusion coefficients, which yielded reasonable energy values at low, middle, and high altitudes across both regions. When we compared the results of the Monte Carlo simulations and the mean particle theory estimates with available observational data, we noted good agreement throughout all studied regions—polar wind, auroral, central polar cap, and cusp. Both the simulation results and the mean particle theory estimates aligned well within the observed data range. We conclude that the velocity diffusion coefficients used in both methods, which produced acceptable consistency with the observations, are appropriate for application in these regions.

For future research, it is essential to conduct further observations in various regions of Earth's magnetosphere. This will facilitate additional comparisons and help identify the most suitable diffusion coefficient, as well as ascertain which method yields the most accurate results in relation to the corresponding observational data.

## 5 Code and data availability

The source code, data, and input files necessary to reproduce the results are available from the authors upon request (barghouthi@staff.alquds.edu).

## 6 Author contribution

First author (Imad Barghouthi) suggested the problem, provided the model (Barghouthi model), discussed the simulation results, and wrote the manuscript. Second author (May Halaika) ran the model (Barghouthi model), obtained the simulation results, and plotted the figures.

## 7 Competing interests

The authors (Barghouthi and Halaika) declare that they do not have any competing interests.

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
