# Peer review of "Comparing Monte Carlo simulations, mean particle theory estimates, and observations of H+ and O+ outflows at high altitudes and latitudes."

_Annales Geophysicae, 2024_

## Author Comment (AC3)

**Response to Reviewer #1**

We would like to thank you for your critical reading of our manuscript and your constructive suggestions and comments.

*Authors: As outlined in the initial paragraph and reiterated throughout your esteemed report, you have suggested removing the mean particle theory model from the manuscript. However, in our investigation, this model yields reliable and valid results, as the diffusion coefficient utilized within it is both suitable and consistent with the coefficient employed in the Monte Carlo model. This alignment indicates that the ion heating process predominates over other forces, as highlighted in various sections of the manuscript. Specifically, the wave-particle interaction effect is shown to be dominant over external forces such as gravity, polarization electric field, and mirror force in certain regions, while it competes with these forces in others. It is acknowledged that, without the correct diffusion coefficient, the mean particle model would indeed become overly simplistic and not advisable for use. Therefore, we advocate for retaining both the model and its associated results within the manuscript.*

**Detailed comments:**

Line 18: …"use velocity and altitude diffusion coefficients…" My understanding is that the authors mean ... use velocity and altitude **dependent** diffusion coefficients. Same in lines 20 and 23.

*Authors: Yes, we mean that "velocity and altitude **dependent** diffusion coefficients", corrected.*

Line 70: remove "…because there are several events at lower altitude.", the reason that the locally observed fields were unlikely to be the source of heating was that they were of too low amplitude. In the subsequent paper (Waara et al. 2011) it was essentially found that this case study was an exception.

*Authors: thank you, we removed the sentence.*

Line 74: I would add: Waara et al. (2011, 2012) provided a statistical study of ion heating and related wave activity. They provided average values of diffusion coefficients…

*Authors: thank you, it has been added.*

Line 78: "They expected the relation…" this sentence is a bit unclear. Looking up the reference I suppose the authors mean that "The electric to magnetic field spectral density ratios were found to be close to what is expected for Alfvén waves."

*Authors: thank you, you are right, we have included your correction.*

Line 88: As suggested elsewhere, I would remove the comparison with the mean particle theory.

*Authors, please, as we mentioned before, we prefer to keep it.*

Equation (8): Spell out explicitly what this equation means, I.e. to my understanding the saturation of the diffusion coefficient due to the finite wave length of the waves. I think it would also be prudent to cite Bouhram et al. here again, as they were first with introducing the finite wavelength effect.

*Authors: we have added the following: "equation (8) indicates the diffusion coefficient is dependent on altitude and velocity. However, as ions are heated and move to higher altitudes, their gyroradius may approach the perpendicular wavelength of the electromagnetic turbulence, and when the ratio ($k_\perp v_\perp / \Omega_i$) exceeds 1, the heating rate becomes self-limited. Bouhram et al. (2004) derived an alternative form for the altitude and velocity dependent diffusion coefficient and interpreted their results in terms of finite wavelength effects."*

Line 164: Once again, I do not think the comparison with an older very simple model adds anything to the paper.

*Authors: again, we prefer to keep it.*

Line 218 section 3.2

This is the interesting part of the paper. Unfortunately it is also the least complete. It is very unclear what the observational data they Cooper with is, and how this study is related to what was already reported in Bargouthi et al (2016). This must be made much clearer. I started reading Barghouti et al. (2016) but it is not really my job to sort this out in a clear fashion. So what I have found is that Nilsson et al. (2013) divided the diffusion coefficients after region. This was extended in Barghouthi et al. (2016) to cover a larger altitude interval as well as different diffusion coefficient combinations, including the maximum and minimum values used in the present paper.

Thus the observations shown are as I understand from Nilsson et al. (2013). The minimum and maximum values around that are from Barghouthi et al. (2016). This should be made clear.

*Authors: we have mentioned that observations are obtained from Nilsson et al (2013) and Barghouthi et al (2016). Also, we present figure 3 and provide the comparison between the results of the models and the corresponding observations in certain mentioned regions.*

Line 233, Figure 3: Minimum and maximum values appear to be interchanged. Highest temperatures are seen for the dotted line, this must correspond to the maximum case and vice versa.

*Authors: sorry for this mistake, we have interchanged them.*

Note that Barghouthi et al. (2016) is missing from the reference list.

*Authors: No, it is in the reference list, line 309.*

---

## Author Comment (AC4)

Response to Reviewer #2

We would like to thank you for your critical reading of our manuscript and we appreciate your respected comments and suggestions.

The article by Barghouthi and Halaika focuses on the physics of ion outflow at the magnetosphere of the Earth and discusses different approaches for modeling this process. By comparing the approaches between them, as well as against observations, the authors conclude on the limitations of each simulation method and the importance of specific parameters which may control the validity and applicability of each method. Generally, the paper is useful and contains original results. Even though there are language and presentation issues, the approach and study concept are straightforward to understand. It is also a potentially useful paper for anyone working on the topic of ion outflow.

*Authors: Thankyou*

On the other hand, the presentation quality of the study is quite low. The authors take too many things for granted (which only experts on outflow may understand), basic introductory materials are missing, e.g. about the outflow theory, what physical processes are involved and how these map to the different part of the equations presented etc. While in some cases references are provided, these are not enough, and I will give more examples below. Furthermore, the study lacks a clear motivation statement. E.g. what is the main reason that this comparison is done? Has this never been done before, is it driven by a necessity to demonstrate the performance and applicability of the Barghouthi model, or is it still unclear which factors (equation terms) control the outflow results? Finally, there are many language issues, e.g. long sentences, sentence parts without articles or written as statements in a conference presentation. I only give selected examples of such language issues below, it is impossible to keep track of them all. I suggest a more careful proofreading.

*Authors: we have published numerous articles, as listed in the references, and many other specialists have contributed extensively to this field of research. Topics such as introductory material, ion heating, ion outflow, Mean Particle Theory, Monte Carlo simulations, and the Barghouthi model have been widely discussed in various publications. In line with our approach, we avoid repeating previously published information and instead provide the key concepts while referring to the relevant references.*

When it comes to the scientific results (mostly Section 3), the main problem I see there is that comparsion between models and observations is under discussed. Despite multiple claims that MC simulations and data agree well, I do spot several key disagreements which need discussion. These missagreements don't falsify the study, but understanding them will also be even more revealing for the theoretical models and their limitations. Furthermore, claims of good/excellent agreements between models and data are not bases on quantitative claims.

*Authors: sometimes, a quantitative comparison isn't possible, so we focus on making qualitative comparisons instead.*

Below I provide selected comments on parts of the paper that justify my summary evaluation above. Overall, I see that all issues are resolvable and the paper can certainly be published after these are resolved. Most comments are minor but adding many minor comments together sums up to a moderate/major revision.

Detailed/specific comments

1) There are many minor or major language issues, e.g. just in the abstract:

Abstract, line 10: Earth magnetosphere -->Earth's magnetosphere
*Authors: corrected*
Abstract, line 11: We present altitude profiles for mean perpendicular◊ add "the" before "mean"
*Authors: done*
**Abstract, line 12:** using Barghouthi model --> using the Barghouthi model
*Authors: done*
**Abstract, lines 15-16:** in which parameter an agreement is obtained?
*Authors: we add, (mean perpendicular energy $W_\perp$ , mean parallel energy $W_\parallel$ , and mean total energy $W_{total}$)*
**Abstract lines 16-17:** What kind of wave particle interaction is referred to here? What external forces refers to? A lot of terminology is used in the abstract, but in a kind of vague way
*Authors: we include, electromagnetic waves, external forces (gravity, polarization electric field, and mirror force)*
**Abstract, line 17:** "produce high energies": I assume high energy particles? Can you indicate numerically what high energy means? what particles are we talking about?
*Authors: high energy $O^+$ and $H^+$ particles, see figures 1, and 2.*
**Abstract, line 18:** "not reasonable": The way this expression is placed in the sentence is not correct and it's unclear what is not reasonable. I suggest to break the long sentence into smaller ones.

*Authors: not reasonable, much higher than observations (Barghouthi, 2008), we break the sentence into sentences.*

**Abstract comments:** Generally, it is not clear what outflow parameters are compared, or what excellent agreement means. There is also no coherence in the text, e.g. "we can claim that the wavelength of the electromagnetic wave existed in those regions": You do not introduce anything about an electromagnetic wave (what is this wave?) . There is no statement of an open question in the beginning sentences of the abstract, it is unclear in the end what exactly is the goal of the study. Within the abstract various terms and concepts are introduced or mentioned which add lots of confusion.

*Authors: we cannot include everything in the abstract, we have mentioned, electromagnetic waves and we provide its wavelength that produce good agreement with observations, we have mentioned the name of the regions, particles, parameters,....*

**Main article:**

**Line 35:** of the ion --> of the ions
*Authors: corrected*
**Line 37-38:** Sentence needs rewrite, maybe break it in several smaller sentences. Also its unclear how Monte-Carlo and diffusion theory are combined, maybe add few words? E.g. "Monte-Carlo simulations are performed using test particles and predefined electromagnetic fields. The way diffusion theory is combined with Monte-Carlo simulations is…". Maybe it is clear for experts in the field but for other readers, numerous unexplained terms and concepts are introduced without background. This will also help understanding text in follow-up paragraphs.

*Authors: We are addressing this to the experts in the field and have included the relevant references along with their key findings. Adding more details would only lengthen the manuscript, and we risk being criticized for repeating information already covered in previous publications.*

**Introduction:** general comment is that by the end of the introduction, no open question is posed. E.g. it is clear that models and theory will be compared, but what is the motivation behind that? Is there still some doubt on which models are best to use? To find the applicability and limitations of each approach? To explore aspects in data that remain unexplained and may require combinations of model? What are the open questions?

Also I need to clarify that since I am not an expert on the topic of the outflow, I would have appreciated some more introductory comments on the topic, that could help readability e.g. in sections like 2.1. For instance, 1-2 sentences on what the polarization electric field is, what are the external forces, what do we refer to when we talk about wave-particle interactions. External forces, for instance, are defined for the first time in line 204, while this could be done in the introduction or section 2.1. WPI is a very broad term. What is the driver/physics of wave activity, what is the topology of the waves (e.g. present at low altitudes, high altitudes?). Without this information, readability of the manuscript would be enhanced and the reach to non- experts can be increased. What is the physics behind the diffusion coefficient (D)? For models not considering WPI, what does D represent?

*Authors: reviewer # 1 mentioned that the manuscript is too long, we cannot include all details, we provide main idea and the major scientific contributions related to the topic of the manuscript.*

**Lines 157-159:** Break this long sentence into several ones, to improve language and readability.

*Authors: done*

**Section 2.2:** Similar to a comment above, can you briefly describe how the Monte-Carlo implementation of your model works? Is this a test-particle approach?

*Authors: yes, it is a test particle approach, it has been described in different publications, Barghouthi 1997, 2008, Barghouthi et al 1998, 2011, 2016, 2016, …*

**Figures 1-4**: Can you add a legend on the Figure? Lines are explained in the caption, but it may be useful to have this figure with a legend in case its used in a review article, presentation etc.

*Authors: We prefer not to do that, as it would make the figures too crowded.*

**Line 207**: Check language – the first sentence after "i.e." is written like it is part of a bulleted list in a presentation. Use simpler writing with smaller sentences. E.g. "This means the velocity diffusion, D(r), coefficient is altitude dependent."

*Authors: yes, corrected*

**Lines 213-217**: Break this long sentence into several ones, to improve language and readability.

*Authors: done*

**Section 2**: I believe it would be useful to have some parametric demonstration that shows at which parameter (or set of parameters) the two approaches (theory and MC simulation) start to deviate. Obviously, the two theories agree well in the auroral region and not in the polar wind region. In line 209-211 it is explained that the poor performance of the mean theory in the polar wind region is due to the effect of the external forces, included in the MC approach only. However external forces are included also in the case of the auroral region. This means that there should be some relation between external forces and the quantification of wave particle interaction, (e.g. a ratio?), across which mean theory becomes irrelevant. Woud it be possible to discuss the results in such a way?

*Authors: We have highlighted in several sections that the wave-particle interaction is especially strong in the auroral region, where it outweighs the external forces. In contrast, in the polar wind region, the wave-particle interaction is weaker, allowing external forces to have a more pronounced impact.*

**Section 3**: This section needs more discussion. There are many claims of excellent or good agreement, but there is no quantitative way to define what that means, besides a non-objective visual comparison of curves in Figures 3 and 4, especially when it comes down to comparison with data. E.g. in the top panels of Fig 3, at altitudes of 6-8 km, the particle energies are in the range of observations, however the shape of the altitude energy profile E(h) is different than what seen in observations. The slope of increasing energy is much steeper in the simulation/theory curves that in the data. It is also unclear why theoretical curves stop below 10 km. In the bottom panels of Fig. 3 the E(h) profiles are more similar, so I would say that the bottom profiles are closer to be "in excellent agreement" than the top ones. The deviations in the top profiles need discussion.

Same applies for the left panel of Figure 4 (steep theory profiles compared to data), whereas the right profiles show better agreement with observations.

*Authors: in the figures 3, and 4, every panel is in different region of earth magnetosphere.*

---

## Author Response (AR1)

Dear Editor,

I hope this message finds you well.

Please accept my sincere apologies for the delayed submission of our revised manuscript. The instability in our country, Palestine, has significantly disrupted our ability to do it early.

Upon reviewing the comments provided by the reviewers, we have addressed each point meticulously. For your convenience, we have included a detailed list specifying the number of lines corresponding to each correction.

We have thoroughly revised the manuscript and believe it is now well-prepared for your consideration.

Thank you for your understanding and continued support.

Best regards,
Imad A. Barghouthi, Professor of physics.

---

## Author Response (AR2)

**Response to Editor Comments**

We thank the reviewer for the thorough reading of our manuscript and for the constructive comments. Below we provide a detailed point-by-point response.

**Comment 1**

*"There are many claims of excellent or good agreement, but there is no quantitative way to define what that means, besides a non-objective visual comparison of curves in Figures 3 and 4"*: please clarify how you define agreement to be good or excellent.

**Response:** In our manuscript, the terms "good" and "excellent" agreement were based on a qualitative visual comparison of the simulation and theory curves with the observational data ranges depicted by the maximum, average, and minimum values in Figures 3 and 4. More specifically:

- We define the agreement as **excellent** when the simulation and theory curves largely overlap with the average observational curves and remain within the observed range across all altitudes.

- We define it as **good** when the curves generally lie within the envelope of the observational maximum and minimum values, even if there are partial deviations from the average trend.

This approach is consistent with qualitative assessments commonly used in comparative studies of ionospheric outflows (e.g., Barghouthi et al. 2012, 2016), given the significant variability inherent in the observational data. and we add a described pargrphe in lines 247 to 254

**Comment 2**

*"In the top panels of Fig 3, at altitudes of 6-8 km, the particle energies are in the range of observations, however the shape of the altitude energy profile E(h) is different than what seen in observations. The slope of increasing energy is much steeper in the simulation/theory curves than in the data"*: please clarify if you only compare absolute energies at selected altitudes or also the slopes of the curves, as requested. How do you explain the different slopes? This is not done in the revision."*

**Comment 3**

*"The deviations in the top profiles need discussion. Same applies for the left panel of Figure 4 (steep theory profiles compared to data), whereas the right profiles show better agreement with observations.

You state that '...the slopes of bottom panels of figure 3 and panels of figure 4 are similar for observations, mean particle theory and Monte Carlo simulations, this is due to the appropriate choice of diffusion coefficients.' However this is just a statement — the top panels of Fig 3 clearly show different altitude/energy slopes compared to data, and this is also partly valid for Fig. 4 (left), especially at higher energies where data-based slopes have break, which does not seem to be developing by the models. Please extend the explanation for these discrepancies, or add more explanation or calculations supporting your statement that discrepancies do not exist."

**Response:** We thank the reviewer for this valuable comment. We have now expanded our discussion to specifically address these discrepancies.

In the revised manuscript, we have now added a paragraph discussing the differences in the slopes of the altitude-energy profiles between our simulation results and the observations. We have added this explanation in the lines 302 to 345